# ARCTIC-3D: automatic retrieval and clustering of interfaces in complexes from 3D structural information

Marco Giulini [1], Rodrigo V. Honorato [1], Jesús L. Rivera[1] & Alexandre M. J. J. Bonvin [1✉]

The formation of a stable complex between proteins lies at the core of a wide variety of biological processes and has been the focus of countless experiments. The huge amount of information contained in the protein structural interactome in the Protein Data Bank can now be used to characterise and classify the existing biological interfaces. We here introduce ARCTIC-3D, a fast and user-friendly data mining and clustering software to retrieve data and rationalise the interface information associated with the protein input data. We demonstrate its use by various examples ranging from showing the increased interaction complexity of eukaryotic proteins, 20% of which on average have more than 3 different interfaces compared to only 10% for prokaryotes, to associating different functions to different interfaces. In the context of modelling biomolecular assemblies, we introduce the concept of "recognition entropy", related to the number of possible interfaces of the components of a protein-protein complex, which we demonstrate to correlate with the modelling difficulty in classical docking approaches. The identified interface clusters can also be used to generate various combinations of interface-specific restraints for integrative modelling. The ARCTIC-3D software is freely available at github.com/haddocking/arctic3d and can be accessed as a web-service at wenmr.science.uu.nl/arctic3d.

[1] Bijvoet Centre for Biomolecular Research, Faculty of Science - Chemistry, Utrecht University, Padualaan 8, 3584 Utrecht, CH, The Netherlands.
✉email: a.m.j.j.bonvin@uu.nl

Protein-protein interactions are of crucial importance in biology, as they are involved in the majority of cellular processes, ranging from signal transduction to cell transport. A key element of a protein-protein complex is the interface between each component of the complex, defined as the set of amino acids of each protein that have at least one heavy atom located within a cutoff distance to the partner (typically 5Å). Biologically speaking, interfaces are typically composed by a subset of amino acids that are crucial for the interaction with the binding partner, while the other residues play a less relevant role[1,2].

Information about interfaces can be extracted from the PDB, but is not always immediate to access and retrieve. The recently released PDBe-graph database[3–7] provides a resource to facilitate the retrieval of such information. In this database, each UNIPROT ID is treated as a node of a network, where the interactions formed with its partners are represented as edges. Notably, the graph-API[5] allows for a fast and programmatic retrieval of such relational data, providing the user with immediate access to the set of interfaces formed by a protein with its partners. This set can consist of multiple interfaces, especially when the protein under study is an interaction hub[8]. Many algorithms attempt at finding similarities between protein-protein interfaces[9–12] and protein-ligand binding sites[13,14] or at analysing and predicting protein interactions[15–17], but none of these focusses on protein-specific interfaces, that is, the sets of residues that are used by a given protein to interact with different partners. We here present an easy-to-use tool to data mine the existing experimental, structural data on protein interfaces.

We introduce ARCTIC-3D (**A**utomatic **R**etrieval and **C**lus-**T**ering of **I**nterfaces in **C**omplexes from **3D** structural information), a software for data mining and clustering the set of available interfaces formed by a reference protein. With the aim of quantitatively distinguishing between different interfaces on a protein structure, we exploit the formal equivalence between interfaces (groups of residues) and coarse-grained mappings[18,19], namely reduced descriptions of proteins in which only a subset of the original atoms (or residues) is retained. Using the mathematical tools developed to quantify the similarities between coarse-grained representations[20], it is possible to assess the similarities between different interfaces.

There are several, potential scientific applications of ARCTIC-3D in structural bioinformatics and, in particular, in the information-driven modelling of molecular systems[21]. We here demonstrate the application of the software with a few examples, ranging from proteome-wide analysis of interfaces to its use for generating interface-specific sets of restraints to guide protein-protein docking.

## Results

### Benchmarking ARCTIC-3D on the Docking Benchmark 5 dataset

As a first test case for ARCTIC-3D we analyse a subset of the Docking Benchmark 5 (BM5) dataset[22], a reference dataset for protein-protein docking. Among the 257 complexes of the dataset, we remove those involving ligands, antibodies and those composed of more than two interacting partners. Note that focusing on two interacting partners is not a limitation of ARCTIC-3D as it is agnostic of the number of interacting partners within a complex when querying the PDBe-KB database. This was done here to simplify the analysis. For each of the 86 remaining complexes, we extract the UNIPROT ID of the two interacting proteins and run ARCTIC-3D on both of them.

Using ARCTIC-3D with default settings, we retrieve an average of 50.9 interfaces and 2.9 interacting surfaces (interface clusters) over the 157 unique UNIPROT IDs in our dataset. For almost all of the 172 proteins constituting the 86 complexes, we find the true

interface with the partner protein among the existing interfaces. The few times (6 cases) in which this does not occur are due to the PDB preprocessing steps, which select a PDB file that does not include coordinates for the amino acids of the aforementioned interface. The only case for which ARCTIC-3D does not retrieve any interface data concerns UNIPROT ID O09130, for which no interface information is available in the PDBe's RESTful API.

For each of these proteins, we looked at how often interfaces formed with the partner UNIPROT ID have been clustered with other interfaces. This analysis is useful to estimate whether an ARCTIC-3D run performed in absence of any knowledge about this protein-protein interaction would still allow to retrieve a reasonable interacting surface. We find that this is the case for 98 entries, namely 57% of the total number of individual proteins.

### The presences of multiple binding surfaces (high recognition entropy) can explain the docking difficulty

In Ref. [22] three ab-initio docking methods (SwarmDock[23], PyDock[24], and ZDOCK[25]) were applied to the 55 new entries of the BM5 dataset. A few quantities, such as interface RMSD (i-RMSD), buried interface area (Δ ASA), and experimental binding free energy (ΔG), were analyzed in order to explain the different docking performances for different complexes. Weak correlations were found between the docking difficulty and the Root Mean Squared Difference of the protein interfaces (i-RMSD) and a combination of buried surface area and binding affinity, but in both cases these were only mildly predictive of the docking success. Another hypothesis could be that the presence of multiple binding surfaces on a protein misleads the docking causing poor performance. This is something that we can easily investigate with ARCTIC-3D.

The intersection between the subset of complexes considered here and described in Ref. [22] amounts to 12 entries. Among those, we excluded from our analysis 3A4S as it involves UNIPROT ID O09130, for which there is no available interface information.

For each complex, we express the docking quality combining the results of the three ab-initio docking software taken from Fig. 1 of Ref. [22] as follows:

$$Q_{\text{docking}} = \sum_{m \in \mathcal{M}} \sum_{t} \frac{Q_{m,t}}{t} \qquad (1)$$

where $\mathcal{M}$ is the set of ab-initio docking methods and $t$ refers to the index of each element in the top $[1, 5, 10, 50, 100]$ array, meaning $t = 1$ when considering the top 1 structure, $t = 2$ for the top 5 and so on. $Q_{m,t}$ is the quality of the best structure produced by method $m$ at the $t$-th element of the top array (1, 2, 3 for acceptable, medium, and high quality models, respectively).

As a measure of the complexity of binding surfaces retrieved by ARCTIC-3D for the two partner proteins we define the following Boltzmann-like entropy, here named "recognition entropy":

$$S_{\text{recognition}} = \ln\left(N_{rec}^{clust} \times N_{lig}^{clust}\right) \qquad (2)$$

which is the natural logarithm of the number of possible combinations of binding surfaces, as given by the product between the number of clusters on the receptor ($N_{rec}^{clust}$) and on the ligand ($N_{lig}^{clust}$). Fig. 1 shows the scatter plot of $S_{\text{recognition}}$ versus $Q_{\text{docking}}$, which shows a clear anti-correlation between the two variables ($r = -0.76$): ab-initio docking software tend to perform consistently well for targets that do not possess many combinations of interface clusters, such as 3CP8 and 3VLB. Instead, the accuracy drops when dealing with complexes whose constituents have multiple binding interfaces, translating into high recognition entropies. In this context, a paradigmatic example is 4M76, a complex for which no ab-initio docking method can find a good solution in the top 100 models ($Q_{docking} = 0$), even though the two

partners do not show any substantial conformational rearrangements upon binding (rigid-body docking category, i-RMSD = 0.43). In this particular complex, the receptor possesses 13 binding surfaces and the ligand 4, and accordingly a high recognition entropy ($S_{recognition}$ = ln(52) = 3.95). This high number of combinations could be the leading cause for the poor performances of all three ab-initio docking software.

**UNIPROT-wide analysis**. In a second benchmarking experiment, we apply ARCTIC-3D to the analysis of a large protein data set, namely the full set of 569213 (as of 9/3/2023) curated proteins present in the UNIPROT Swiss-Prot[26] database. Running ARCTIC-3D with default parameters on this huge amount of UNIPROT IDs required 11.08 CPU hours on a 50 AMD EPYC

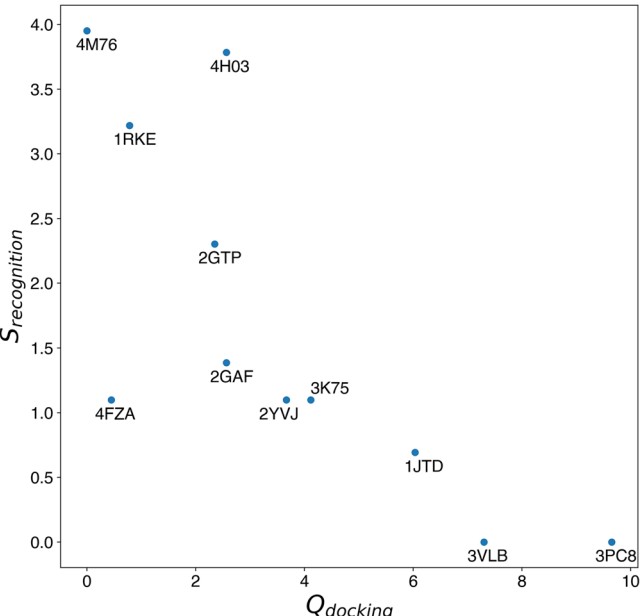

**Fig. 1 Scatter plot of $Q_{docking}$ (x-axis, Eq. (1)) against $S_{recognition}$ (y-axis, Eq. (2)) for 11 entries of the updated bm5 dataset.** The ab-initio docking performances seem to depend on the complexity of the protein interactome.

7451 processor. The speed performance is highly dependent on the protein of interest, as a UNIPROT ID with no interface information (95.9% of the proteins, 87.0% of the overall execution time) simply amounts to a call to the PDBe graph API[5], while an *interaction hub* with a long sequence results in longer execution time, the majority of it being due to the download of the PDB files associated to each UNIPROT ID. For example, the ARCTIC-3D run for *SARS-CoV-2 spike glycoprotein* (UNIPROT ID P0DTC2) took more than half an hour to complete. A comprehensive analysis of the speed performance of different stages of ARCTIC-3D is available in the Supplementary Material section 1 (Supplementary Material Figs. 1–2 and Supplementary Material Table 1).

ARCTIC-3D could retrieve information for 23446 UNIPROT IDs (4.12% of the total number of entries analyzed). 59.73% of the considered proteins have a single interacting surface, while 14.75% of them display more than three. Fig. 2 shows the histogram of the number of binding surfaces for Homo sapiens and the four main taxonomic superkingdoms, namely *Eukaryotes*, *Bacteria*, *Archaea*, and *Viruses*. From the plot we can observe how the histograms for *Eukaryotes* and *Homo sapiens* display a slower decrease while going from left to right, that is, moving towards proteins with a considerable number of interacting surfaces. The value displayed on each histogram represents the fraction of proteins that possess more than 3 interface clusters. This behaviour may have multiple possible explanations: first, human and eukaryotic proteins tend to be longer than non-eukaryotic proteins[27], therefore simply having more space for accommodating multiple binding surfaces. Second, eukaryotic proteins have been investigated more in detail than their counterparts[28], and the number of annotated protein-protein interfaces may be higher for them.

**Data-driven docking**. It is possible to generate docking restraints from two ARCTIC-3D runs by means of the `arctic3d-restraints` command line interface (see Methods section). In this section, as an example, we apply this idea to a real docking scenario, using 1AVX[29] from the BM5 dataset[22]. The free form of the components of the complex map to two PDB files: 1QQU[30], chain A (crystal structure of porcine beta trypsin) and 1BA7[29], chain B (crystal structure of Kunitz-type soybean trypsin inhibitor). This is supposedly a relatively easy docking, as the interface $C_\alpha$ atoms show a sub-angstrom RMSD (0.47 Å) between the

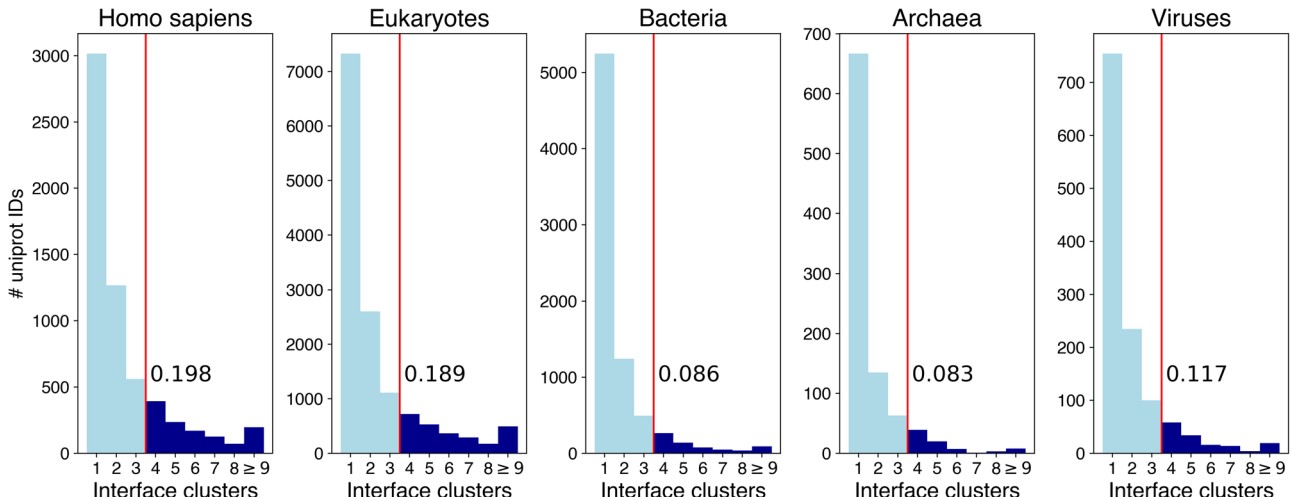

**Fig. 2 Histogram of the number of interface clusters for Homo sapiens and the four taxonomic superkingdoms.** The number reported in each plot next to the vertical red line represents the fraction of proteins of each category with more than 3 interacting surfaces. The ninth bin of each histogram refers to proteins with 9 or more clusters.

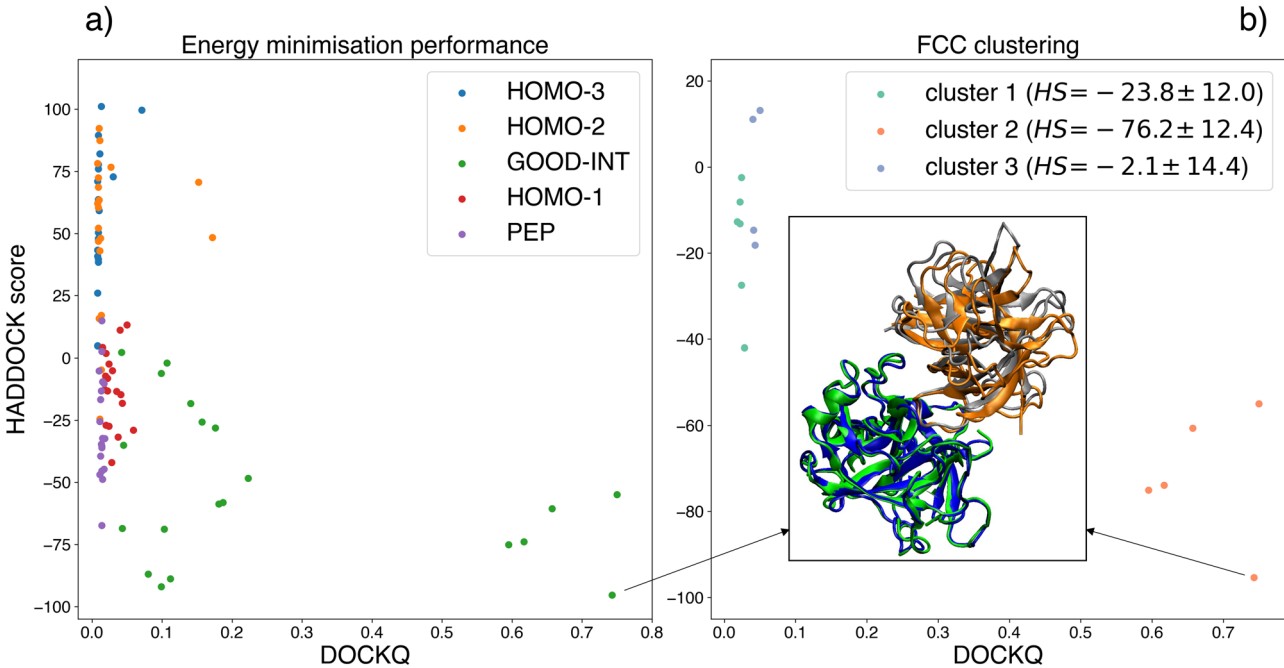

**Fig. 3 Accuracy of HADDOCK3 generated models (DOCKQ score) plotted against the HADDOCK score.** The colour-coding indicates the interface restraint combination used to drive the docking: GOOD-INT corresponds to the identified interface overlapping with the true interface for this complex, PEP to the identified interface with the heterochiral peptide, HOMO-1, HOMO-2, and HOMO-3 to the three homodimeric binding surfaces found for P00761. **a** Single model statistics for the 100 generated models after final energy minimisation. **b** Cluster-based statistics showing the model part of the three identified clusters. The best-scoring model (receptor in green, ligand in silver) is shown superimposed onto the target complex (receptor in blue, ligand in orange) (i-RMSD = 1.175 Å, DOCKQ = 0.743). Cluster scores and standard deviations are reported in the legend. Those are calculated on the top 4 models of each cluster.

bound and the unbound structure. The complex falls within the rigid-body category of the BM5 dataset.

ARCTIC-3D was run on the two UNIPROT IDs corresponding to 1QQU and 1BA7, namely P00761 (*Sus scrufa Trypsin*) and P01070 (*Soybean Trypsin inhibitor A*), excluding from the analysis the structure of the complex, 1AVX, and also all interfaces formed by P00761 (resp. P01070) with P01070 (resp. P00761), as in a real-case scenario this kind of information would typically be unavailable. When running ARCTIC-3D for P01070 we impose the reference PDB file to be 1BA7. Only one interface is retrieved, namely the one formed in 6O1F[31] with *Homo sapiens Tryptase alpha/beta-1* (UNIPROT ID Q15661). The retrieved list of residues shows a substantial similarity with the real P00761-P01070 interface present in 1AVX.

For P00761 and choosing 1QQU as reference PDB, ARCTIC-3D returns 43 interfaces, which cluster into 5 binding surfaces. The recognition entropy of this ARCTIC-3D run is then equal to $S_{recognition} = \ln(1 \times 5) = 1.61$. Among the retrieved surfaces, three correspond to homodimeric (P00761-P00761) interfaces, while the other two concern different categories of heterodimers: one is a secondary binding surface formed by the trypsin with a hetero-chiral peptide (PDB id 1V6D[32]), while the other (a cluster of 25 different interfaces) covers the standard trypsin binding surface.

Assuming complete ignorance about the location of the true binding surface, restraints were generated with `arctic3d-restraints` with the default probability threshold $P^{thr} = 0.3$ for all combinations of the P00761 binding surfaces with the single binding surface identified for P01070.

This set of five interaction ambiguous restraints was input in HADDOCK3, the new modular version of HADDOCK[33], with a fast, low-sampling docking workflow, composed by the following steps:

1. **rigid-body energy minimisation**, in which only 100 solutions are sampled, namely only 20 for each set of input restraints;
2. **flexible refinement** of all 100 models;
3. **final energy minimisation**;
4. **Fraction of Common Contacts clustering**;[12] using a 0.6 similarity cutoff and requiring a minimum of 4 models per cluster;
5. **Cluster-based scoring** with the default scoring function of HADDOCK[34] which consist of a linear combination of intermolecular van der Waals and electrostatic energies using the OPLS[35] force field, an empirical desolvation energy term[36] and the restraint energy.

The underlying assumption behind this reduced computational protocol is that the presence of good information in part the input data (one of the five interaction restraints) should allow one to retrieve acceptable docking solutions even when limiting the sampling. In this case this proved to be correct, as HADDOCK identifies 5 medium quality docking poses out of the 100 sampled. Fig. 3 shows the correlation between the quality of the docking models, expressed with the DOCKQ metric[37] and the HAD-DOCK score. The first medium quality model (DOCKQ = 0.743) ranks at position 1 (Fig. 3a). Cluster-based analysis clearly identifies the near-native cluster as top-ranking one (Fig. 3b). This HADDOCK3 run took 13 minutes and 12 seconds on 20 AMD EPYC 7451 CPU cores. The limited sampling at the rigid body energy minimisation allows for a faster execution of the workflow with respect to the standard HADDOCK recipe.

In this case, HADDOCK produces a complex whose accuracy is comparable to that obtained using Alphafold2-multimer[38,39], whose top-ranked model has a DOCKQ score equal to 0.831. This complex was however present in the training set of Alphafold2.

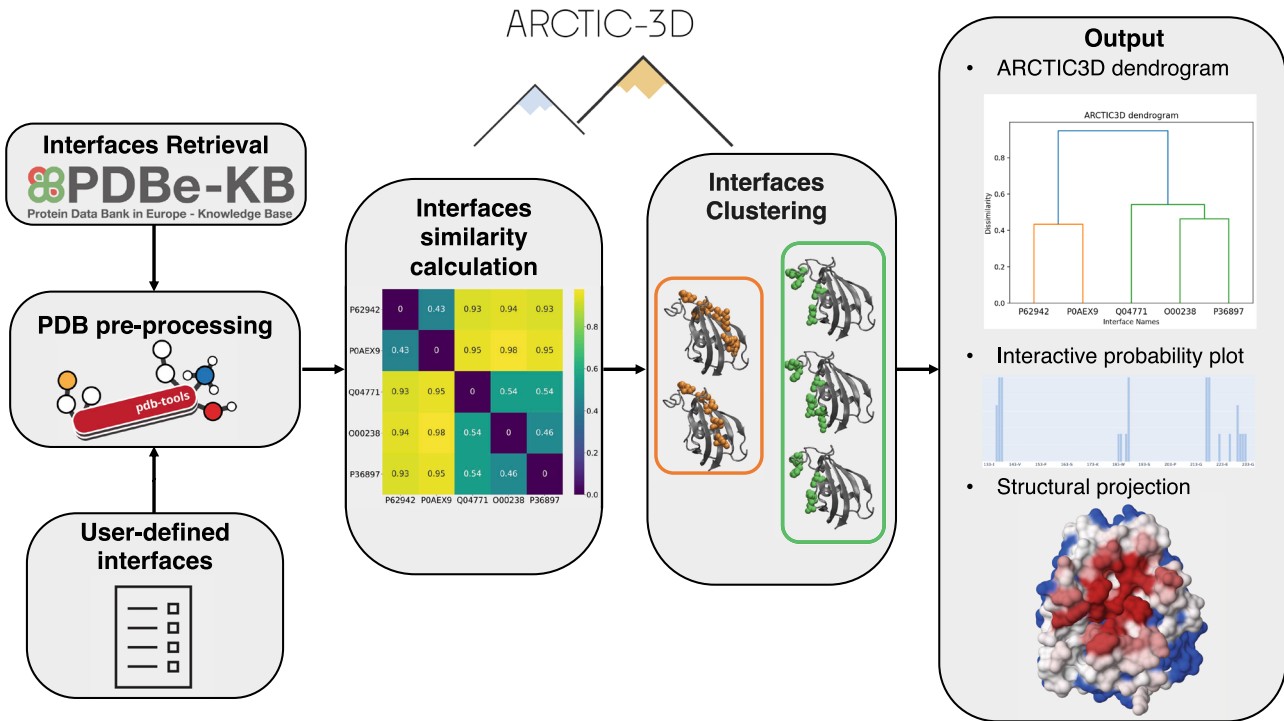

**Fig. 4 Schematic representation of a typical ARCTIC-3D workflow.** In the first stage of the execution (left panels), interfaces are either retrieved from the PDBe's RESTful API[5] or read from a user-provided input file. Those interfaces are then projected over a cleaned PDB 3D structure, making it possible to calculate a similarity matrix between them (centre-left panel) and, finally, to cluster them in separate binding surfaces (centre-right panel). In the latest stage of the execution (right panel), the results are provided to the user by means of PDB files in which the binding surface information is encoded into the $\beta$ factor field and interactive plots.

## Discussion

In this work we have presented ARCTIC-3D, a novel tool for the retrieval and classification of protein interfaces from the PDB database. Once provided with protein input data, ARCTIC-3D queries the PDBe-graph database[6] through its graph-API[5], extracting the available information about the protein of interest. Subsequently, the formal equivalence between a protein interface and a coarse-grained reduced representation (mapping) is exploited to derive a notion of similarity between different interfaces. This is then used to identify the different binding surfaces of the structure under investigation.

Applications of ARCTIC-3D to a subset of the Docking Benchmark 5 dataset[22] and to the Uniprot Swiss-Prot[26] database prove the tool to be a reliable and unsupervised source of information when it comes to the analysis of annotated protein interfaces. We have also shown, that, for protein-protein complexes, next to the amount of conformational changes taking place upon binding, the number of potential binding surfaces can explain the modelling difficulty. For this we have introduced the concept of recognition entropy. The interface-specific restraints generated by ARCTIC-3D, whose use was demonstrated with HADDOCK, should be useful to any modelling software that can make use of such information for model generation and/or scoring.

Another useful application of ARCTIC-3D concerns the analysis of binding surfaces according to the proteins that interact with them to search for subcellular location, biological process, and molecular function of these partners. It provides an unbiased, and computationally inexpensive method to assess whether one of these factors is related to the considered binding surface.

In conclusion, ARCTIC-3D, available both as a standalone code and user-friendly web service, offers an intuitive and simple protocol to fetch and rationalise protein interface information, with the aim of facilitating the understanding and visualisation of the available binding surfaces.

## Methods

This section details the functioning of the program, from interface mining to clustering, and provides some usage examples. A schematic representation of the full ARCTIC-3D workflow is provided in Fig. 4.

**Data mining of interfaces.** The software can accept three categories of input, namely a sequence, a UNIPROT ID (recommended), or a PDB file.

When a sequence or a PDB file are provided, we determine the associated UNIPROT ID by means of a BLASTP search. ARCTIC-3D then performs a HTTP request to the PDBe's RESTful API[5] to gather all the available interaction information. This data is parsed, according to different parameters. When the input is a PDB file, the user has the freedom to skip this step by submitting an `interface-file` with a list of curated interfaces, which might be the results of experiments, computational modelling, or previous ARCTIC-3D runs.

In the following step ARCTIC-3D exploits again the PDBe's RESTful API[5] to get the PDB file to be used for the subsequent geometric calculations (if it was not provided in input). The API provides a list of structures ranked by sequence coverage and resolution (undefined for NMR structures). ARCTIC-3D downloads the corresponding mmcif files[40] and converts those to PDB format, renumbering the amino acids according to the UNIPROT numbering scheme so as to ensure consistency between interfaces and structures. These PDB files are then processed and cleaned using pdb-tools 2.5.0[41]. As not all interface residues might be present in all PDB files, for the clustering analysis we select the

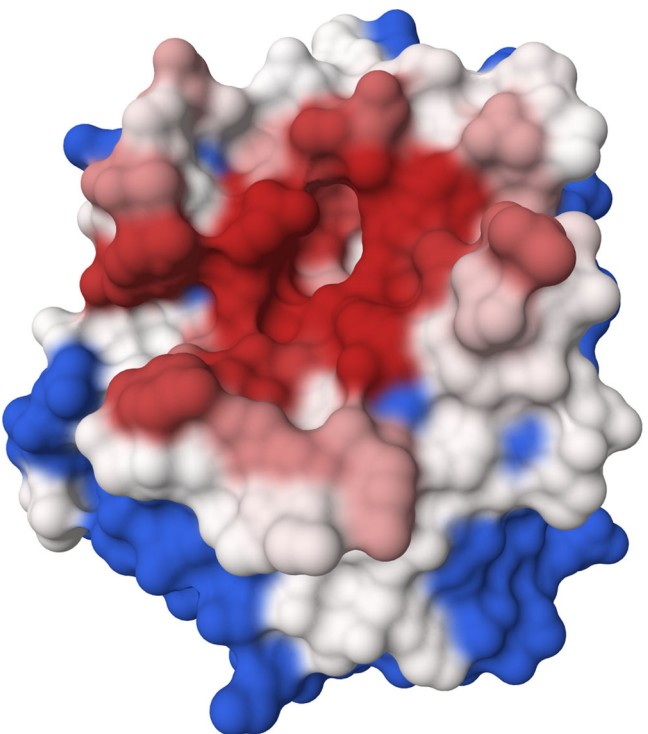

**Fig. 5 Example of ARCTIC-3D output structure for cluster 1 of UNIPROT ID P00760.** Residues in red are those with a high probability (1.0 or very close) to be in the binding surface. Residues in white show intermediate-to-low values of probability (see Eq. (10)), while amino acids in blue are never observed to be in this interface cluster. Image produced with Molstar[49].

PDB file that retains the highest number of interfaces. The user can speed-up this operation by forcing the algorithm to use a specific chain of a specific PDB file (using the `pdb-to-use` and `chain-to-use` parameters). By default, an interface is retained if at least 70% of its residues are present in the PDB structure under analysis. This threshold can be modified by the user through the `int-cov-cutoff` parameter.

**Interface similarity and clustering**. Having pre-processed the PDB file and the set of filtered interfaces we can proceed to determine their mutual similarity and, ultimately, cluster them in binding surfaces. It is possible (see Ref. [20]) to associate a subset of protein atoms to a vector $\phi$ in an abstract space, namely the Hilbert space of square-integrable real functions $L_2(\mathbb{R}^3)$.

An interface $I$ can be considered a subset of atoms of a protein and projected onto $L_2(\mathbb{R}^3)$. It is then useful to measure the properties that characterise the interface itself, such as its norm $\mathcal{E}(I)$, and its similarity with other interfaces by calculating the scalar product $\langle \phi_I, \phi_J \rangle$. These quantities are calculated as Ref. [20]:

$$\mathcal{E}(I) = \sum_{i,j=1}^{n} J_{ij} \chi_{I,i} \chi_{I,j} \tag{3}$$

$$\langle \phi_I, \phi_J \rangle = \sum_{i,j=1}^{n} J_{ij} \chi_{I,i} \chi_{J,j} \tag{4}$$

$$\chi_{I,i} = \begin{cases} 1 & \text{if atom } i \in \text{ interface I,} \\ 0 & \text{if atom } i \notin \text{ interface I.} \end{cases} \tag{5}$$

where the sums run over all the considered atoms of a proteins (here only the $C_\alpha$ atoms for simplicity) and $J_{ij}$ is a gaussian

coupling between atoms $i$ and $j$, which depends on their pairwise distance $r_{ij}$:

$$J_{ij}(r_{ij}) = e^{-r_{ij}^2/4\sigma^2}, \tag{6}$$

The gaussian width $\sigma$ is here set to half the distance between two consecutive $C_\alpha$ atoms (1.9 Å), as in Ref. [20].

Following these definitions, the distance between two interfaces $I$ and $J$ can be calculated as:

$$\mathcal{D}(I,J) = \left( \mathcal{E}(I) + \mathcal{E}(J) - 2\langle \phi_I, \phi_J \rangle \right)^{\frac{1}{2}}$$
$$= \left( \sum_{i,j=1}^{n} J_{ij}\chi_{I,i}\chi_{I,j} + \sum_{i,j=1}^{n} J_{ij}\chi_{J,i}\chi_{J,j} + -2\sum_{i,j=1}^{n} J_{ij}\chi_{I,i}\chi_{J,j} \right)^{\frac{1}{2}}. \tag{7}$$

In this case $I$ corresponds to the set of $C_\alpha$ atoms belonging to the amino acids of the protein that constitute the interface.

When dealing with real interfaces, though, using the distance metric defined in Eq. (7) might not be the wisest choice as it heavily depends on the number of residues present in interfaces $I$ and $J$. Even when these two interfaces span the same region on the protein surface, their distance might be non-negligible when they differ in the number of residues forming each interface.

As a measure of similarity between two interfaces, we therefore propose to consider instead the angle between two interfaces, which can be easily calculated[20] as:

$$\cos\theta_{I,J} = \frac{\langle \phi_I, \phi_J \rangle}{(\mathcal{E}(I)\mathcal{E}(J))^{\frac{1}{2}}}, \tag{8}$$

and, in particular, the sine of such angle:

$$\sin\theta_{I,J} = \sqrt{1 - \cos^2\theta_{I,J}} \tag{9}$$

The sine of the angle is a robust quantity to observe, as a very high value corresponds to complete orthogonality between interfaces, i.e., the interfaces occupy two completely different regions of the protein surface. On the other hand, $\sin\theta_{I,J} \sim 0$ corresponds to the situation of absolute parallelism between interfaces, which is obtained when $I$ and $J$ span the same region on the protein surface. We provide an example of the behaviour of the distance (Eq. (7)) and the sine of the angle (Eq. (9)) for various interfaces in Supplementary Material section 2 (see also Supplementary Material Fig. 3).

Once the similarity matrix between all interface is computed based on Eq. (9), we use agglomerative hierarchical clustering[42] to retrieve the interface clusters. This clustering method was selected because of its mathematical rigour and its ability to handle directly distance matrices. The resulting clusters can be thought of as binding surfaces, obtained by combining together slightly different sets of interacting amino acids. By default, the average linkage prescription[43] is used to generate the hierarchy of clusters (dendrogram, see an example in Fig. 4 and in Supplementary Material Fig. 4). The default threshold used to stop the hierarchical grouping procedure (i.e. for clustering) is 0.866, corresponding to an angle of 60 degrees (see Supplementary Material section 3 and Supplementary Material Fig. 5). Both `linkage` and `threshold` are input parameters of ARCTIC-3D and can be changed by the user.

**Output example and interpretation**. We provide here a brief description of the output produced by the software using UNI-PROT ID P00760, namely *Bos Taurus Serine Protease 1*, as an example.

The data mining stage of the algorithm retrieves 228 interfaces formed by this protein, which are saved and can be re-used (for example as an interface file). The PDB validation process selects PDB ID 4XOJ, chain A, as the valid entity containing 3D coordinates for residues covering the highest number of interfaces

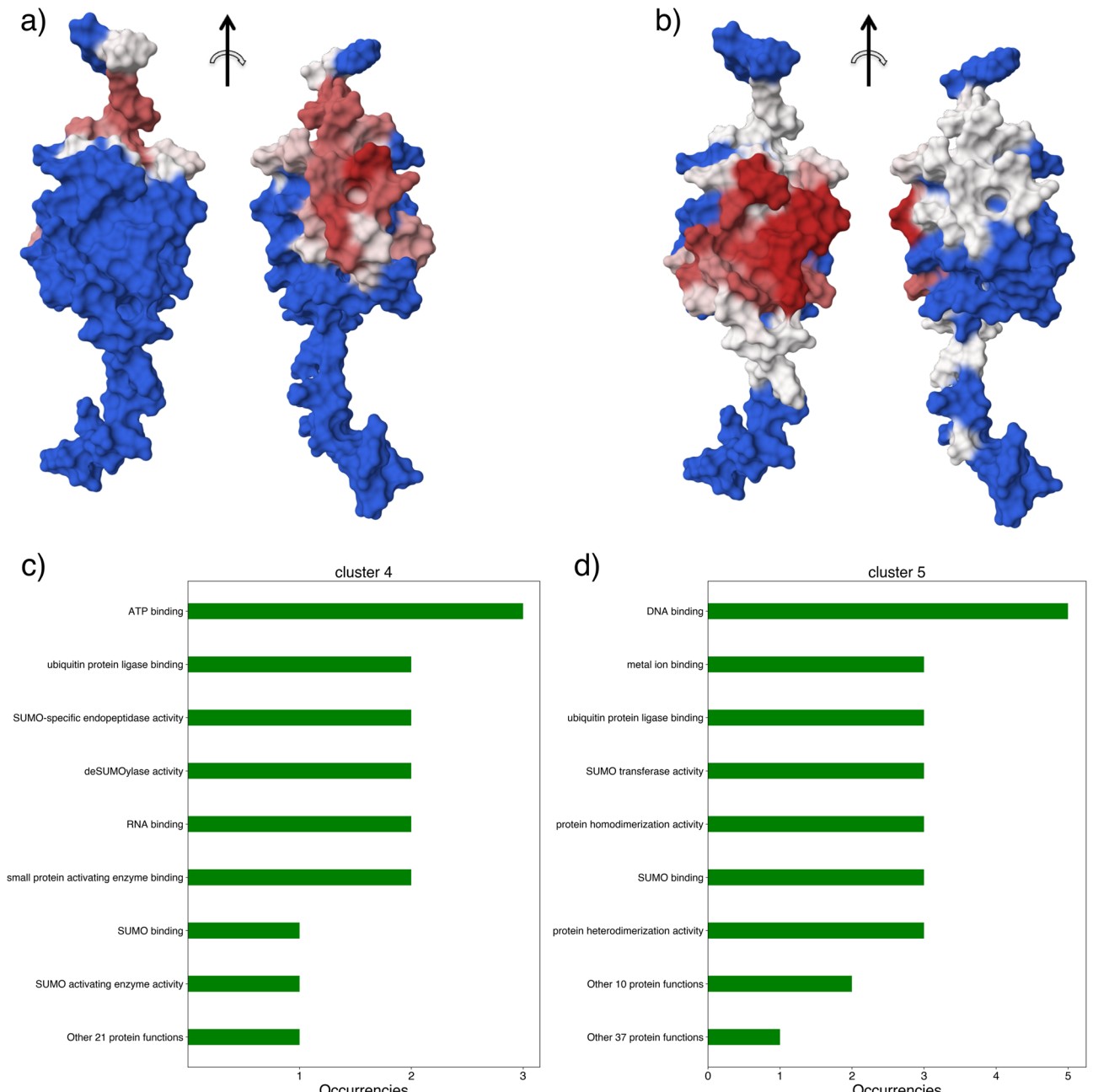

**Fig. 6 Results of the application of** `arctic3d-localise` **to two binding surfaces of P63165 (SUMO-1), namely cluster 4 and cluster 5. a, b** The two different binding sites corresponding to cluster 4 and 5, respectively, following the same colour scale reported in Fig. 5. **c, d** The output of `arctic3d-localise` for those two clusters: for cluster 4, there is no preferred biological function among the partners, while the second interacting region shows a clear preference for DNA-binding proteins.

(all of them in this case). These interfaces are compared and the similarity matrix between them is computed and saved. Then, clustering is performed, producing a total of 7 distinct binding surfaces using default parameters. The corresponding dendrogram is plotted (see Supplementary Material Fig. 4).

For each cluster/binding surface $\mathcal{K}$, ARCTIC-3D outputs the following items:

- the set of interfaces belonging to $\mathcal{K}$, each one characterised by a name and a set of residues;
- the probability of each residue $i$ to belong to the $\mathcal{K}$ binding surface, $P_{i,K}$, calculated as the fraction of times the residue is observed within the cluster;

- a PDB file with the aforementioned probabilities embedded in the $\beta$ factor column, according to the following formula:

$$\beta = \begin{cases} 50.0\,(1 + P_{i,K}) & \text{if residue } i \in \text{ cluster K,} \\ 0.0 & \text{otherwise} \end{cases}$$

(10)

This $\beta$ factor scale is introduced to make the interacting residues more evident in common molecular visualisation software.

Fig. 5 shows a graphical rendering of such PDB for cluster 1. An interactive plotly[44] plot shows the different cluster

probabilities over the canonical protein sequence. Examples are available at wenmr.science.uu.nl/arctic3d/example and in the Supplementary Material section 4, where we show the ability of ARCTIC-3D to discriminate between alternative interfaces belonging to the same protein (see Supplementary Material Fig. 6).

**ARCTIC-3D-resclust, ARCTIC-3D-localise, and ARCTIC-3D-restraints**. In addition to the main command line interface (CLI), we introduce three other commands that can be useful when manipulating ARCTIC-3D results, and more generally, interface information.

*arctic3d-resclust*. In a variety of common situations one might have a set of possibly interacting residues, obtained either from experiments and/or bioinformatic predictions. In these cases, this list of residues may effectively correspond to more than one interface. arctic3d-resclust clusters a residue list over an input PDB structure. Since in this case we are considering single residues and not interfaces (collections of residues), we use the $C_\alpha - C_\alpha$ Euclidean distance to compute the distance matrix.

Default values for linkage strategy, cutoff distance and clustering criterion parameters are "average", 15 Å, and "distance", respectively. All these parameters can be easily adjusted by users. An example scenario for arctic3d-resclust is presented in the Supplementary Material section 5 (Supplementary Material Table 2 and Supplementary Material Fig. 7).

*arctic3d-localise with meaningful data*. A well-established idea in the field of protein-protein interactions is that similar structural interfaces may allow a protein to bind to similar partners[9]. Such partners may not only share similarities in structure, but also in biological function or subcellular location.

It is therefore interesting to investigate the results of ARCTIC-3D runs from this perspective to search for similarities among the partner proteins binding at each interacting surface. The arctic3d-localise command is devoted to such task: given the result of a standard ARCTIC-3D run, it loops over the interacting partners retrieving information about their subcellular location, their function, or the biological process they are involved in. This is made possible by calls to the UNIPROT[26] and QUICKGO[45] databases. Once such information has been retrieved, the existing clustering performed by ARCTIC-3D is used to divide the partners and their function over the various binding surfaces.

As an example application of arctic3d-localise we analysed the *Homo sapiens Small ubiquitin-related modifier 1* (SUMO-1, UNIPROT ID P63165). ARCTIC-3D retrieves 94 interfaces from the PDB, divided into six binding surfaces. Four of them are quite small and not highly populated, while the remaining two (cluster 4 and cluster 5, see Fig. 6) contain 41 and 46 interfaces, respectively.

Using arctic3d-localise we can discriminate these binding surfaces according to the biological function of the interacting partners. Fig. 6d shows how 5 of the 9 (curated) partners of cluster 5 contain the DNA binding label. In short, the binding surface characterised by cluster 5 is typically used in interactions with proteins that are able to bind DNA, while the other interacting surface (cluster 4) is never found in contact with DNA binding domains. This is consistent with literature data showing how residues belonging to the interacting surface 5 of SUMO-1 (such as K37, K39, H43, and K46, see Fig. 6c) are key for the interaction with SUMO interaction motifs on DNA-binding proteins[46]. arctic3d-localise can thus be used to quickly elucidate if any correlation exists between binding surfaces and protein function, subcellular localisation, and biological processes.

*arctic3d-restraints*. Interface information retrieved by means of ARCTIC-3D can be used to drive the modelling of the complex. The various binding surfaces found from the interacting partners (as defined by ARCTIC-3D) can be used for scoring generated models, imposing a penalty whenever these are not present at the interface, or more directly to drive the modelling process by defining restraints between the interfaces as done in HADDOCK or in AlphaFold2[38,39]-based methods such as AlphaLink[47] or ColabDock[48]. For use in HADDOCK the interface information is translated into ambiguous interactions restraints. Instead of combining all interfaces into one set of restraints, arctic3d-restraints allows to generate different sets of restraints for each combination of binding surfaces. In doing that, ARCTIC-3D does not select by default all the residues of a binding region, but only those amino acids that are consistently present in the region, namely those that are observed to be there more often than a certain, pre-defined frequency. This number, called $P^{thr}$ is set by default to 0.3 and can be modified by the user.

**Reporting summary**. Further information on research design is available in the Nature Portfolio Reporting Summary linked to this article.

## Data availability

The data generated in this study are available at zenodo.org/record/8131701.

## Code availability

The ARCTIC-3D code, together with various usage scenario examples, is freely available at github.com/haddocking/arctic3d.

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

## Acknowledgements
The authors acknowledge Sameer Velankar and the whole PDBe team for the invaluable resources provided by the PDBe database and PDBe graph-API and their responsiveness to questions about the use of the API. This project has received funding from the European Union Horizon 2020, projects BioExcel (823830 and 101093290) and EGI-ACE (101017567), and from the Netherlands e-Science Center (027.020.G13).

## Author contributions
A.M.J.J.B supervised the project. M.G. designed the algorithm. M.G. developed the software and performed all the experiments. R.V.H developed the web interface. J.L.R performed the initial exploratory work and wrote the initial scripts. A.M.J.J.B and M.G. wrote the manuscript.

## Competing interests
The authors declare no competing interests.

## Additional information

**Peer review information** : *Communications Biology* thanks the anonymous reviewers for their contribution to the peer review of this work. Primary Handling Editors: [EBM name(s)] and [Internal Editor name(s)]. A peer review file is available.

