## [Peer Review File · Communications Biology]

Reviewers' comments:

Reviewer #1 (Remarks to the Author):

The work presents a method for analysing variable interfaces of monomeric protein components as they interact in different protein complexes. The method presented extracts interfaces (the definition of what an interface is is not clear) and compares the atoms in these using thresholding (named Gaussian couplings). The couplings are compared through the angle between them, serving as a kind of alignment of the different thresholds (to my understanding, this is not very clear). The paper continues with an analysis of different interface properties and the potential for improved rigid docking. Finally, an analysis of interface differences in Uniprot across different kingdoms is provided, which is the most interesting analysis presented here.

Overall: Interesting concept that may be very useful going into the future with a rapid increase in the number of complexes with available structure. Substantial work to prove that the method actually captures different interfaces and the utility over other available methods is needed. Analyses of known interface differences across kingdoms and proteins from previous work should be discussed as well.

Specific comments:

Main issues:

What cutoff was used for the analyses here, i.e. how do you define an interface? This should be made clear.

What is lacking is a comparison with other methods such as the 3Di alphabet from FoldSeek and some type of ground truth set of what different interfaces really are. I suggest creating a set of variable interfaces through manual annotation to analyse how many of these are captured. Now, an intuition for the clustering procedure is provided but no proof that this actually captures meaningful differences is presented. The evaluation on the Docking benchmark 5 dataset does not consider the retention of alternative interfaces which is the objective of the software presented here.

Other issues:

How does the selection of the cutoff threshold impact the analysis? Is there a better/alternative way to define interfaces than using cutoffs such as the 5Å suggested here?

It is stated that: "An interface is discarded if less than 70% of residues are present in the structure under analysis.". What makes 70% a good cutoff? Is it possible that the interfaces are affected even if only 20% of the structure is missing?

The authors mention that "Even when these two interfaces span the same region on the protein surface, their distance might be non-negligible when they differ in the number of residues forming each interface." and instead consider the angle between interfaces. Please provide some intuition to the reader why the difference in number of residues matter and why the angle is better in such a case (i.e. illustrative examples, not only equations). It is also unclear how the angle function performs at small and large dissimilarities. If the angle is e.g. 80 vs 90 degrees, how will this affect the similarity compared to at 10 vs 20 degrees? Provide an analysis of the correspondence in capturing different interfaces of using angles/not and how these functions vary (number of residues vs angle and number of captured interfaces).

In the interactive example you provide: wenmr.science.uu.nl/arctic3d/example, you reuse the colours (although there are only a few clusters). This makes it hard to distinguish the clusters. Is it possible to also plot the clusters on the 3D structure and visualise this interactively? This would make it easier for users to identify the different interface regions.

Figure 1 legend: It is not clear what is depicted in the "interfaces similarity calculation" (perhaps refer to this) and the text is too small to read in that part of the figure.

It would be interesting to see the overlap in interface similarity between the different kingdoms. Are all interface clusters of Bacteria also in Eukarya or what are the fractions? This question is important to answer to understand the complexity of the interface space across the kingdoms of life and their varieties. Perhaps Venn diagrams can be added.

How fast is the method? Do a benchmark using a selection of complexes from different kingdoms. Now, a very rough approximation is provided and a statement about that an interaction hub will take longer. This is quite obvious and the authors should instead time the search for different proteins and hub sizes.

The analysis of the impact on docking is not that meaningful in the post-AlphaFold age. The rigid docking methods that are analysed require bound monomeric structures and even if these are provided, the performance is very low according to recent benchmarks (close to 0% success rate for rigid docking methods, <https://www.nature.com/articles/s41467-022-28865-w>). If this is to be included, a comparison with AlphaFold(AF)-multimer and AF+restraints such as in <https://www.biorxiv.org/content/10.1101/2023.07.04.547599v1> and/or <https://www.nature.com/articles/s41587-023-01704-z> should be added.

In Figure 6, the relationship between the HADDOCK score and the DockQ score is not very convincing and the number of datapoints seem to be too few to properly assess this. Please see the above comment about AF additions and extend this analysis to more complexes.

Reviewer #2 (Remarks to the Author):

This paper introduces ARCTIC-3D, a fast and user-friendly data mining and clustering software to retrieve data and rationalise the interface information associated with the protein input data, including sequence, UNIPROT ID and PDB file. The authors demonstrate a few example use-cases for the application of ARCTIC-3D, ranging from proteome-wide analysis of interfaces to its use for generating interface-specific sets for restraints to guide protein-protein docking. ARCTIC-3D finally provides a user-friendly web service. However, I have a few questions.

- (1) In the title, "Clustering" should be replaced by "Clustering".
- (2) In METHODS, why is the interface discarded when less than 70% of residues are present in the structure under analysis?
- (3) This paper uses agglomerative hierarchical clustering to retrieve the interface clusters. Whether other clustering algorithms can achieve better results?
- (4) In this paper, experiments are lack control groups.

Reviewer #3 (Remarks to the Author):

The authors designed a method ARCTIC-3D to analyze the interfaces in protein complexes from 3D structural information, and developed a corresponding software. The work is of interest and importance. Here, I have several questions:

- (1) Are there any related studies? I think a discussion is required talk about it.
- (2) As there are many clustering algorithms, why do the authors use a hierarchical clustering strategy?
- (3) How should we analyze those complexes composed of more than two interacting partners?

We thank all reviewers for their time and effort in reviewing our manuscript. In the following we provide detailed answers to their comments. The modified parts are highlighted in red in the manuscript.

Reviewer #1:

The work presents a method for analysing variable interfaces of monomeric protein components as they interact in different protein complexes. The method presented extracts interfaces (the definition of what an interface is is not clear) and compares the atoms in these using thresholding (named Gaussian couplings). The couplings are compared through the angle between them, serving as a kind of alignment of the different thresholds (to my understanding, this is not very clear). The paper continues with an analysis of different interface properties and the potential for improved rigid docking. Finally, an analysis of interface differences in Uniprot across different kingdoms is provided, which is the most interesting analysis presented here.

Overall: Interesting concept that may be very useful going into the future with a rapid increase in the number of complexes with available structure. Substantial work to prove that the method actually captures different interfaces and the utility over other available methods is needed. Analyses of known interface differences across kingdoms and proteins from previous work should be discussed as well.

Specific comments:

Main issues:

1) What cutoff was used for the analyses here, i.e. how do you define an interface? This should be made clear.

That is indeed a relevant point, and we apologise the reviewer for not making it clear in the first version of the manuscript. In the default execution scenario of ARCTIC-3D, interfaces are retrieved from the PDBe Graph API (https://www.ebi.ac.uk/pdbe/graph-api/pdbe_doc/), which uses a cutoff of 5Å. As such we do not calculate the interfaces ourselves, but use the pre-calculated data from the PDBe database. Note that the 5Å cutoff is a well-accepted cutoff to identify interfaces which is standardly used for example in CAPRI, the critical assessment of prediction interactions.

However, a user-defined list of interfaces can be provided to the software (thanks to the `interface_file` parameter) and, while creating such file, a user can define his/her own cutoff, thus allowing flexibility for scenarios in which an alternative way to define interfaces is needed. In this work, however, we trust the quality of the PDBe data.

This has been clarified in the manuscript.

2) What is lacking is a comparison with other methods such as the 3Di alphabet from FoldSeek and some type of ground truth set of what different interfaces really are. I suggest creating a set of variable interfaces through manual annotation to analyse how many of these are captured.

There seems to be some confusion here related to the previous point. We do not predict or calculate any interfaces, but rather extract the ground truth directly from the PDB-KB database. Those interfaces are defined purely based on the 3D coordinates information from the complexes deposited in the PDB. Of course, users can also provide their own definition

of the interface but testing all possible ways of defining a custom interface is outside the scope of this work.

3) Now, an intuition for the clustering procedure is provided but no proof that this actually captures meaningful differences is presented.

To address this point, we added two new sections in the Supplementary Material titled “Understanding and visualizing angles and distances between interfaces” and “Retention of alternative interfaces: an example” in which we show two examples that show the capability of the software to capture meaningful differences between interfaces.

4) The evaluation on the Docking benchmark 5 dataset does not consider the retention of alternative interfaces which is the objective of the software presented here.

Again, there is some confusion as ARCTIC-3D considers all interfaces documented in the PDBe-KB databases, i.e. also all alternatives to those present in BM5. This is at the core for example of the concept of recognition entropy which we introduce here.

Other issues:

5) How does the selection of the cutoff threshold impact the analysis? Is there a better/alternative way to define interfaces than using cutoffs such as the 5Å suggested here?

5Å intermolecular distance is a well accepted cutoff in the field, which is being used within the CAPRI community to define contacts. And again, we do not recalculate the interfaces but take the information directly from the PDBe-KB database.

6) It is stated that: “An interface is discarded if less than 70% of residues are present in the structure under analysis.”. What makes 70% a good cutoff? Is it possible that the interfaces are affected even if only 20% of the structure is missing?

This percentage is used when selecting the entry from the PDB database that contains the highest number of acceptable interfaces. When mapping the interfaces onto the structure, we want to select the representative PDB file which covers the most interfaces. Setting this cutoff too high might discard too many interfaces, and too low would not allow to properly visualize and cluster the interfaces as too many residues would be missing from the 3D structure. 70% is a good compromise. But to give more freedom to users we added a parameter (**int_cov_cutoff**) in the input (both command line and server) that allows to define it (see Pull Request <https://github.com/haddocking/arctic3d/pull/342>).

7) The authors mention that “Even when these two interfaces span the same region on the protein surface, their distance might be non-negligible when they differ in the number of residues forming each interface.” and instead consider the angle between interfaces. Please provide some intuition to the reader why the difference in number of residues matter and why the angle is better in such a case (i.e. illustrative examples, not only equations).

We agree with the reviewer and added a new section in the supplementary information named “Understanding and visualizing angles and distances between interfaces” devoted to this explanation (and referenced in the Methods section). We show an example that clarifies why the angle is better than the distance in preserving the biological similarity between interfaces.

8) It is also unclear how the angle function performs at small and large dissimilarities. If the angle is e.g. 80 vs 90 degrees, how will this affect the similarity compared to at 10 vs 20 degrees? Provide an analysis of the correspondence in capturing different interfaces of using angles/not and how these functions vary (number of residues vs angle and number of captured interfaces).

The reviewer suggested a very interesting analysis. We added a section in the SI (titled "On the clustering cutoff threshold") in which we perform the suggested comparison. In clustering there is not a unique threshold value that is valid for all needs but combining the output dendrogram (an example in the Suppl. Material) and the threshold input parameter ARCTIC-3D users can easily fine-tune the clustering procedure.

As for the difference, the same difference (for example 10 degrees) in the angle will cause significant changes in the clustering at low angles, while it will be less impactful at big angles. This has been mentioned in the analysis. It is also important to note that the clustering is done based on the sine of the angle, and not the angle itself.

9) In the interactive example you provide: wenmr.science.uu.nl/arctic3d/example, you reuse the colours (although there are only a few clusters). This makes it hard to distinguish the clusters. Is it possible to also plot the clusters on the 3D structure and visualise this interactively? This would make it easier for users to identify the different interface regions.

We thank the reviewer for this insightful suggestion. We corrected the plot generation to have markedly different colors (see Pull Request <https://github.com/haddocking/arctic3d/pull/334>). For each cluster, residues are plotted on the 3D structure of the protein with a color code related to the probability (e.g. frequency) of observing them in the cluster.

An updated version of the example is available at <https://wenmr.science.uu.nl/arctic3d/example>

10) Figure 1 legend: It is not clear what is depicted in the "interfaces similarity calculation" (perhaps refer to this) and the text is too small to read in that part of the figure.

We agree with the reviewer on this, and we adjusted the figure and the corresponding legend accordingly.

11) It would be interesting to see the overlap in interface similarity between the different kingdoms. Are all interface clusters of Bacteria also in Eukarya or what are the fractions? This question is important to answer to understand the complexity of the interface space across the kingdoms of life and their varieties. Perhaps Venn diagrams can be added.

This is a very interesting suggestion which will require a significant amount of work. The current manuscript describes ARCTIC-3D with the interface characteristics of various organism as application example. We intend to follow up on this suggestion in future work.

12) How fast is the method? Do a benchmark using a selection of complexes from different kingdoms. Now, a very rough approximation is provided and a statement about that an interaction hub will take longer. This is quite obvious and the authors should instead time the search for different proteins and hub sizes.

We had already reported the overall execution time for the UniProt benchmark. We are now providing a more comprehensive analysis in the SI (SI Figure 5). Briefly, we show how the PDB retrieval step is, at least given our bandwidth and the current PDB download API provided by the PDBe, the bottleneck of the method, accounting for more than 60% of the

total execution time (which remains below 10 CPU seconds for the vast majority (72%) of the considered proteins).

13) *The analysis of the impact on docking is not that meaningful in the post-AlphaFold age. The rigid docking methods that are analysed require bound monomeric structures and even if these are provided, the performance is very low according to recent benchmarks (close to 0% success rate for rigid docking methods, <https://www.nature.com/articles/s41467-022-28865-w>). If this is to be included, a comparison with AlphaFold(AF)-multimer and AF+restraints such as in <https://www.biorxiv.org/content/10.1101/2023.07.04.547599v1> and/or <https://www.nature.com/articles/s41587-023-01704-z> should be added.*

We have to disagree with the reviewer here as there are still plenty of problems AI (and AlphaFold can not handle properly), one example of which being antibody-antigen complexes. Of course, we are well aware of the successes of AlphaFold (for protein-protein and protein-peptide modelling), but the question of docking difficulty linked to the recognition entropy we introduce is also relevant for other types of complexes (e.g. including nucleic acids). Further, AlphaFold-multimer has been trained including the BM5 complexes and an independent test is therefore impossible. This manuscript is not about docking, but analysis of interfaces from the existing structural knowledge of the PDB.

The ARCTIC-3D interface-specific restraint docking (performed from unbound structures) is just an illustration of the method. AlphaFold-Multimers does give a perfect solution for this complex (but it was part of the training set). Note that the Arctic-3D derived restraints could also be used in AF2 or any other software which could use such information to bias the modelling.

To make this clear added a note about AF2 in the relevant section and cite it.

14) *In Figure 6, the relationship between the HADDOCK score and the DockQ score is not very convincing and the number of datapoints seem to be too few to properly assess this. Please see the above comment about AF additions and extend this analysis to more complexes.*

Again, this is not a docking manuscript. Figure 6 only shows an application example with very limited sampling.

Reviewer #2 (Remarks to the Author):

This paper introduces ARCTIC-3D, a fast and user-friendly data mining and clustering software to retrieve data and rationalise the interface information associated with the protein input data, including sequence, UNIPROT ID and PDB file. The authors demonstrate a few example use-cases for the application of ARCTIC-3D, ranging from proteome-wide analysis of interfaces to its use for generating interface-specific sets for restraints to guide protein-protein docking. ARCTIC-3D finally provides a user-friendly web service. However, I have a few questions.

(1) *In the title, "ClusTering" should be replaced by "Clustering".*

This was done to explain the acronym. We have changed it in the title and added the explanation of the abbreviation in the introduction.

(2) *In METHODS, why is the interface discarded when less than 70% of residues are present in the structure under analysis?*

This point was also raised by reviewer 1 and answered above. Please refer to point 6 in our answers to the first reviewer.

(3) This paper uses agglomerative hierarchical clustering to retrieve the interface clusters. Whether other clustering algorithms can achieve better results?

The reviewer is right in pointing out this aspect, and we apologise for not discussing it in depth in the first version of the manuscript. We selected the agglomerative hierarchical clustering procedure because of its mathematical rigor and complete determinism, which ensures exact reproducibility without having to specify seeds or other randomness-related parameters. Furthermore, hierarchical clustering is capable of handling directly distance matrices in input, without needing an explicit feature matrix, which here would be impossible to calculate without a necessarily approximate data projection (Principal Component Analysis, Multi Dimensional Scaling, ...). K-means clustering cannot be applied here for this reason.

A more technical reason for choosing hierarchical clustering is that an excellent python implementation of the method is available in *scipy*, while most clustering algorithms are only implemented within the *scikit-learn* package, which is quite heavy (50 MB including dependencies).

We added a short note at the end of the Methods section summarizing this.

(4) In this paper, experiments are lack control groups.

We here assume that the reviewer is referring to some sort of measure that quantifies the correctness of the software's output. The software does not perform any sort of prediction, but rather takes information coming from experimental data (the PDB website) and creates clusters of real, existing interfaces. As such the concept of control groups seems irrelevant to us, but we might be misunderstanding the reviewer's comment.

Reviewer #3

Remarks to the Author:

The authors designed a method ARCTIC-3D to analyze the interfaces in protein complexes from 3D structural information, and developed a corresponding software. The work is of interest and importance.

We sincerely thank the anonymous reviewer for the positive feedback on our manuscript.

Here, I have several questions:

(1) Are there any related studies? I think a discussion is required talk about it.

The reviewer is right in highlighting this point. Many methods exist that aim at analysing or clustering protein-protein interfaces (from any model provided). Several works focused on the related task of interface residue prediction. As an example, the PeSto software (*Krapp et al, Nature Communications*) analyses the whole proteome (like ARCTIC-3D) but focusing on the prediction rather than the retrieval and clustering problem. According to our knowledge, though, no method able to gather and quantitatively compare existing protein-specific interfaces from the PDB-KB database has been proposed before. We added a discussion about this point in the introduction.

(2) As there are many clustering algorithms, why do the authors use a hierarchical clustering strategy?

Please refer to our answer to reviewer 2 on this point (comment number three)

(3) How should we analyze those complexes composed of more than two interacting partners?

ARCTIC-3D is in principle agnostic of the number of components of a complex, as it performs a protein-centric analysis of all interfaces of that protein present in the PDB-KB database. This analysis can of course be repeated for all components of a complex.

REVIEWERS' COMMENTS:

Reviewer #1 (Remarks to the Author):

All comments have been assessed to a sufficient degree. I would like to have seen a comparison to other recent methods, especially to the 3Di alphabet of FoldSeek as I think this would have been interesting. Likely, a more fine-grained approach is needed to capture a larger variety of differences, although the one presented here is functional.